# Global, regional, and country-level cost of leptospirosis due to loss of productivity in humans

**Suneth Agampodi** [1,2]*, **Sajaan Gunarathna**[3], **Jung-Seok Lee**[1], **Jean-Louis Excler**[1]

**1** International Vaccine Institute, Seoul, Republic of Korea, **2** Department of Internal Medicine, School of Medicine, Yale University, New Heaven, Connecticut, United States of America, **3** Department of Community Medicine, Faculty of Medicine and Allied Sciences, Rajarata University of Sri Lanka, Mihintale, Sri Lanka

* Suneth.Agampodi@ivi.int

## Abstract

### Background

Leptospirosis, a prevalent zoonotic disease with One Health priority and a disease of poverty, lacks global economic burden estimates. This study aims to determine the global, regional, and country-level cost of leptospirosis due to loss of productivity.

### Methodology/principal findings

The cost of leptospirosis due to loss of productivity (referred to as productivity cost hereafter) was estimated by converting the disability-adjusted life years (DALYs) lost due to leptospirosis to a monetary value using the per capita gross domestic product (GDP). The country-specific DALYs lost were obtained from the global burden of leptospirosis study published previously. Non-health GDP per capita (GDP- per capita health expenditure) was used for the cost conversion of DALYs. Country-specific GDP and health expenditure data were obtained from the World Bank data repositories. Estimates were done using both nominal and international dollars.

The estimated global productivity cost of leptospirosis in 2019 was Int$ 29.3 billion, with low and high estimates ranging from Int$ 11.6 billion to 52.3 billion. China (Int$ 4.8 billion), India (Int$ 4.6 billion), Indonesia (Int$ 2.8 billion), Sri Lanka (Int$ 2.1 billion), and the United States (Int$ 1.3 billion) had the highest productivity cost due to leptospirosis. Eight out of 10 countries with the highest burden were in the Asia-Pacific region. In addition, lower-middle-income countries had an annual productivity cost of Int$ 13.8 billion, indicating that the disease is poverty-related.

### Conclusion

Although significant, the cost estimate due to loss of productivity is merely a fraction of the overall economic burden of this disease, which also includes other direct, indirect, and intangible costs. The existing partial estimates of the different components of economic cost suggest a profound economic burden that demands the inclusion of leptospirosis in the global

**Data Availability Statement:** All relevant data are within the manuscript and its Supporting information files.

**Funding:** The author(s) received no specific funding for this work.

**Competing interests:** The authors have declared that no competing interests exist.

health agenda for comprehensive disease control and prevention efforts, including vaccine development.

## Author summary

Leptospirosis is a widespread disease that affects humans and animals and is mostly affecting individuals living in resource-poor settings, in particular in tropical and subtropical regions. To understand the economic impact of this disease, we estimated the cost of this disease, assuming that all deaths and disabilities due to the disease will lead to a loss of human productivity. The country-level estimates were done using the published data from the World Bank on per capita gross domestic product (GDP) after removing the per capita health expenditure from the GDP. The estimated average annual cost of leptospirosis due to loss of productivity was $29.3 billion in 2019, which could be as high as $52.3 billion. The countries with the highest costs were China ($4.8 billion), India ($4.6 billion), Indonesia ($2.8 billion), Sri Lanka ($2.1 billion), and the United States ($1.3 billion). Most countries with the highest costs were in the Asia-Pacific region, and lower-middle-income countries had an annual cost of 13.8 billion US dollars. The estimate of lost productivity represents, however, a fraction of the overall economic burden caused by leptospirosis, highlighting the need for immediate action to control and eliminate the disease and its inclusion in the global health agenda.

## Introduction

Leptospirosis is one of the most widespread zoonotic diseases globally [1,2]. Human leptospirosis is caused by more than 35 species of pathogenic and intermediate *Leptospira* [3] transmitted primarily through environmental sources. Rodents (in particular rats) and farm animals, including cattle, pigs, sheep, and other domestic (dogs) and peri-domestic mammals, are the maintenance hosts for the human leptospirosis [4]. Humans are accidental hosts, and the infection mostly causes mild flu-like symptoms [5]. Even though early oral antibiotic treatment is often effective, late presentation during the immune phase of the disease usually leads to complications. Around 10–15% of the cases may develop severe complications with multiorgan failures requiring specialized care. The well-known Weil's disease [6], reported more than a century ago with severe hepatic and renal dysfunctions, represents only a fraction of severe complications. Fatal complications without jaundice, such as severe pulmonary hemorrhages and myocarditis, are increasingly reported in recent literature [7,8].

Since the Global Burden of Disease (GBD) study [8] does not include leptospirosis as a separate entity, the comparative burden of leptospirosis compared to other NTDs were not available in the literature until recently. In 2015, Costa et al. [9] estimated the leptospirosis incidence worldwide using country-level data available from national surveillance programs and estimated 1.03 million cases (95% CI 434,000–1,750,000) and 58,900 deaths (95% CI 23,800–95,900) due to leptospirosis annually [9]. However, these figures are believed to be vastly underestimated due to limited awareness of the disease, non-specific clinical presentations, limited availability of diagnostic facilities, and poor reliable surveillance systems [10]. In addition, the available diagnostic tests are suboptimal, with the best available tests missing almost 40% of the cases [11]. Based on the 2015 GBD, Torgerson et al. have estimated that approximately 2.90 million (Uncertainty Interval ER 1.25–4.54 million) disability-adjusted life

years (DALYs) are lost per annum globally due to leptospirosis [12]. They emphasized that the burden of leptospirosis is about 70% of the global burden of cholera estimated by the 2010 GBD and much more than several other diseases listed as neglected tropical diseases.

Leptospirosis is a disease of poverty, even in high-income settings [13,14]. An increase per capita household income of 1$ per day was associated with > 11% risk reduction in leptospira antibody positivity rate in Brazil, a country with a high burden of leptospirosis [2]. Several other analytical studies also show a direct and significant association of leptospirosis with the economic status [2,15]. While the risk and impact are mostly borne by urban and rural poor communities, the economic impact of leptospirosis at the country level is also estimated to be extremely high. In Brazil, the estimated financial impact of leptospirosis amounted to $ 4.33 million in non-earned wages, while hospitalization costs were estimated to be around $ 157,000 in 2010 [16]. The outbreak of leptospirosis associated with 2011 floods and mudslides in Rio de Janeiro was estimated to cost up to $ 100,800 [17]. In Manila, The Philippines, the total economic value of preventing leptospirosis was estimated to be $ 124.97 million per annum [18], which was 1.13% of metro Manila's GDP. Saizan et al. estimated that preventing leptospirosis in Kelantan, Malaysia, would save around $ 106.7 to 315 million per annum [19]. In 2021, the economic impact of leptospirosis in New Zealand was estimated to be around $ 4.42 (95% CI: 2.04–8.62) million per annum [20].

Despite being identified as the most widespread zoonotic disease, a disease with One Health priority, and a disease of poverty, estimates of the global burden of leptospirosis in economic terms represented an important gap. Our study aims at estimating the global, regional, and country-level cost of leptospirosis attributed to the loss of productivity in humans.

## Methods

DALYs were used for the first time to estimate the burden of diseases in the 1993 World Health Report [21]. It was defined as "A unit used for measuring both the global burden of diseases and the effectiveness of health interventions, as indicated by reductions in the disease burden." The underlying rationale, assumptions, and technical basis were published later in 1994 [22]. The GBD estimates were done for the year 1990 for the first time using DALYs [23]. Despite early rationale criticisms [24], DALYs have become the most widely used indicator for disease burden assessment.

In 2001, the World Health Organization (WHO) Commission on Macroeconomics and Health [25] converted the DALYs lost to a monetary value using the Gross Domestic Product (GDP) per capita with the human capital approach proposed by Weisbrod in 1961 [26]. This method was easy to understand because it assumed that each DALY lost had a cost equivalent to the GDP per capita of the country. Therefore, the total cost of lost productivity due to disease could be calculated by multiplying the total number of annual DALYs lost by the GDP per capita of the country. This proposed approach was subsequently used to evaluate the economic value of DALYs lost due to violence [27] and cancer [28]. The concept is that a loss of health (in DALYs) of a person in each country will lead to a loss of economic contribution to this country as compared to the contribution of a healthy person, defining the average GDP per capita of the country. The economic value was calculated by multiplying country-specific GDP by disease-specific DALYs for each country. The WHO guideline on Identifying the Economic Consequences of Disease and Injury report proposed that health expenditure should not be included in these calculations [29]. Hence, in subsequent studies [30–36], non-health GDP or net GDP was used to evaluate the economic value of DALYs lost. Non-health GDP is "the component of a country's GDP that excludes total health expenditure." It determines value costs unrelated to direct health spending resulting from illness or death. More recently, the

Box 1. Calculation of leptospirosis-related gross domestic product loss calculation

$$\mathbf{NHGDPPC}_{country} = GDPPC_{country} - PCHE_{country}$$

$$\textbf{\textit{Lepto\_NHGDPLoss}}_{country} = \left[ \mathrm{NHGDPPC}_{country} \right] \times \left[ Lepto_{DALYs_{country}} \right]$$

$$\textit{Global LeptoNHGDPLoss} = Lepto\_NHGDPLoss_{country\_1} + Lepto\_NHGDPLoss_{country_2} + Lepto\_NHGDPLoss_{country\_x}$$

$\mathbf{NHGDPPC}_{country}$ = Non-health GDP per capita of the country

$\mathbf{GDPPC}_{country}$ = GDP per capita in the country

$\mathbf{PCHE}_{country}$ = Per capita health expenditure in the country

$\mathbf{Lepto\_NHGDPLoss}_{country}$ = Leptospirosis-related gross domestic product loss in the country

$\mathbf{Lepto\_DALYs}_{country}$ = DALYs loss due to leptospirosis in the particular country

WHO's productivity loss due to illnesses in Africa [37] used the same methodology to evaluate the productivity loss due to DALYs lost. In our study, we used a similar approach in calculating the productivity cost of DALYs lost due to leptospirosis (Box 1).

The most recent data for GDP per capita and per capita health expenditure by country was obtained from the World Bank [38]. We used the World Bank 2019 data for these estimates as the COVID-19 pandemic period may have drastically shifted these numbers. GDP per capita based on purchasing power parity international dollars (Int$), as well as nominal GDP, were used for these estimates to provide a better understanding of productivity loss. The Int$, also known as the Geary-Khamis dollar, is a hypothetical unit of currency that has the same purchasing power parity that the US dollar has in the United States at a given point in time. It allows for comparing the purchasing power parity between countries. In other words, Int$ follows the law of one price, reflecting people's living standards comparably across countries. It is widely employed by the International Comparison Program and the World Bank for international economic analysis and comparisons. Whenever 2019 data are missing, we use the most recent data available. For missing details on per capita health expenditure, the Institute of Health Metrics and Evaluation country-specific estimates were used [39]. Leptospirosis-related estimated DALYs lost by country and region were obtained from the Torgerson study [12]. Since calculating uncertainty intervals using the conventional 95% confidence limits is impossible, we calculated the lower and upper estimates for the productivity cost using the 95% confidence limits for DALYs estimated previously using the same equation as above. The results are presented as the 'estimate range' (ER). The productivity cost of DALYs lost due to leptospirosis was calculated at the country, WHO region, and global levels. Upper and lower estimates for regional and global levels were calculated as a sum of each country's upper and lower estimates.

## Results

All data needed for the productivity cost estimates was available for 196 countries and territories. The global annual productivity lost due to leptospirosis based on 2019 GDP was Int$ 29.3 billion with low and high estimates of Int$ 11.6 and Int$ 52.3 billion, respectively. This was based on a nominal $ 13.9 (ER 5.5–24.8 billion) loss of productivity using actual GDP. China (Int$ 4.8 billion ER 1.9–8.3 billion) and India (Int$ 4.6 ER 2.0–8.2 billion) had the highest economic burden due to DALYs lost. Indonesia followed these countries with an annual productivity cost of Int$ 2.8 billion (ER Int$ 1.1–5.1 billion). The only other country with more than Int$ 2 billion annual loss was Sri Lanka (Int$ 2.1 billion, ER .8–4.0 billion). The top 20 countries with productivity cost by Int$ and US$ are displayed in Fig 1, and all the countries with estimates are listed in S1 Table.

The global distribution of productivity cost due to leptospirosis (Fig 2) shows that the highest economic burden is clustered in the Asia-Pacific region.

High-burden countries other than China, India, and Sri Lanka in the region include Thailand, Vietnam, Indonesia, and Malaysia, each country having a productivity cost of more than Int$ 500 billion due to leptospirosis. The only country outside the region with a productivity

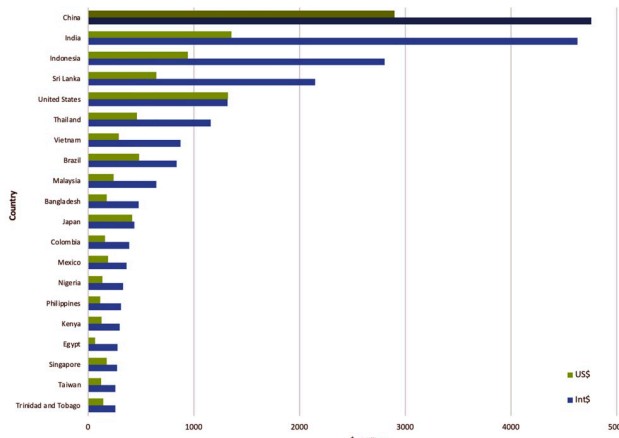

**Fig 1. Top 20 countries with the highest cost of leptospirosis due to loss of productivity.** All figures are given in international dollars (Int$). Y axis numbers are in millions.

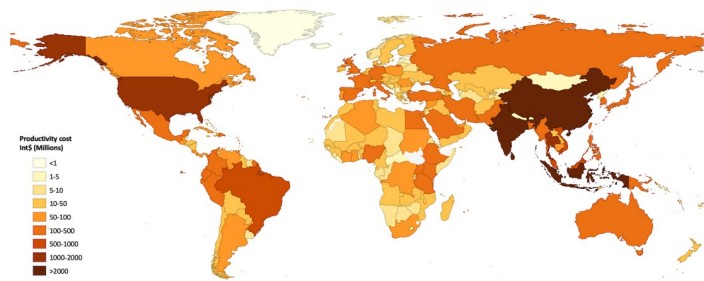

**Fig 2. The global distribution of the cost of leptospirosis due to loss of productivity.** Baselayer maps were downloaded from the opensource data https://public.opendatasoft.com/explore/dataset/world-administrative-boundaries/export/.

**Table 1. Global distribution of estimated annual productivity cost of leptospirosis due to DALYs lost by WHO region.**

| WHO Regions | Productivity cost (Int$)[1] | Upper and lower estimates (Int$) | | Productivity cost (US$) | Upper and lower estimates (US$) | |
|---|---|---|---|---|---|---|
| African | 1,937,996,407 | 739,094,557 | -3,337,325,680 | 775,863,188 | 295,728,661 | -1,335,295,766 |
| Eastern Mediterranean | 1,284,118,102 | 516,267,661 | -2,135,490,689 | 399,211,603 | 159,188,707 | -669,281,417 |
| European | 1,937,421,097 | 802,695,456 | -3,426,223,900 | 1,286,012,864 | 537,356,028 | -2,270,458,608 |
| Region of the Americas | 4,430,758,412 | 1,733,096,813 | -7,969,553,002 | 3,023,193,450 | 1,195,628,678 | -5,404,925,390 |
| South East Asia | 11,385,519,772 | 4,573,376,496 | -20,400,597,987 | 3,640,451,762 | 1,460,752,247 | -6,517,502,271 |
| Western Pacific | 8,289,452,560 | 3,233,631,946 | -15,061,334,503 | 4,788,401,223 | 1,868,051,914 | -8,632,437,265 |

**Table 2. Global distribution of estimated annual productivity cost of leptospirosis due to DALYs lost, by income level.**

| Income Level (# of countries) | Productivity cost (Int$) | Upper and lower estimates (Int$) | | Productivity cost (US$) | Upper and lower estimates (US$) | |
|---|---|---|---|---|---|---|
| High (63) | 4,991,838,706 | 2,003,990,113 | -9,039,553,914 | 4,029,419,816 | 1,621,044,478 | -7,250,787,445 |
| Upper Middle (53) | 9,710,429,956 | 3,806,465,983 | -17,203,489,291 | 5,176,081,465 | 2,031,428,599 | -9,142,054,649 |
| Lower Middle (53) | 13,794,730,030 | 5,493,031,496 | -24,744,093,891 | 4,428,531,402 | 1,757,442,046 | -7,946,895,527 |
| Low (26) | 768,267,660 | 294,675,337 | -1,343,388,665 | 279,101,408 | 106,791,113 | -490,163,096 |

cost of over a billion dollars was the USA, with an estimated Int$ 1.4 loss per year (ER Int$ .5 —2.5 billion).

For the WHO regions, the highest economic burdens were in the South-East Asia region and the Western Pacific region (Table 1), accounting for 38.9% and 28.3% of the global burden, respectively.

The productivity loss by income level (Table 2) shows that 53 lower middle-income countries and territories listed had the highest burden, with a total of more than Int$ 13 billion, accounting for almost half (47.1%) of the global burden.

## Discussion

The productivity cost or the human capital approach was not widely used in evaluating the economic impact on health until recently. Previously, researchers have looked at the different components of the economic burden of leptospirosis. The social gradient and socio-economic status measured in monetary terms have been investigated so far as a risk [2,40], out-of-pocket expenditure [41], cost of hospitalization [16], opportunity cost [18], social cost [17], and cost of prevention [19] related to human leptospirosis in different geographical locations. In addition to the human leptospirosis-related costs, leptospirosis in domestic animals also contributed to a massive economic cost [42–44]. The reproductive loss in farm animals related to leptospirosis and the cost of prevention and control of animal leptospirosis could be much higher than the cost of human leptospirosis. For example, in New Zealand, the production loss cost in beef cattle, sheep, and deer was estimated to be around US$ 7.92 million, while the median vaccination cost in cattle, sheep, and deer was US$ 6.15 million. In contrast, the human leptospirosis-associated estimated cost was US$ 4.42 million [20]. Yet, comprehensive cost of illness studies and global economic burden estimates for human leptospirosis or leptospirosis as a One Health burden is not available.

In the backdrop of the lack of any global economic burden estimates on leptospirosis, our conservative estimates show that 2.9 million DALYs accrued due to leptospirosis annually could have accounted for a loss of productivity of Int$ 29.3 billion in countries included in this analysis. The upper estimate using the upper value of 95% confidence limits of DALYs shows

over Int$ 52.3 billion annual loss. The highest economic impact was on the Asia-Pacific, with China, India, Indonesia, Sri Lanka, and Thailand having the maximum burden. This result is not unexpected because the high disease burden in Sri Lanka, Thailand, and Indonesia is well documented [45–47]. The high productive cost of leptospirosis in Sri Lanka, compared to larger countries such as China and India, may seem unrealistic based solely on incidence rates. However, it is important to recognize that Sri Lanka's proactive approach towards physician awareness, clinical detection, and notification of leptospirosis was triggered by a large outbreak in 2008 [48]. This outbreak led to increased disease awareness, surveillance, and reporting in the country [49]. As a result of these efforts, Sri Lanka may provide a more accurate representation of the true incidence and economic cost of leptospirosis compared to other countries that may not have had a similar outbreak or response. Therefore, it is reasonable to assume that the burden of leptospirosis in many other resource-poor settings and countries, in particular in Africa, may be grossly underestimated due to limited awareness of the disease, poor sensitivity of the diagnostic test, lack of disease surveillance and reporting, as well as limited resources for prevention and control efforts. This underestimation has already been reported in India [50].

As a disease with a heavy productivity cost burden, leptospirosis requires more global attention on control and preventive strategies. The published systematic reviews on leptospirosis chemoprophylaxis show no beneficial effect [51]. Because the severe complications and deaths are typically in the immune phase, the role of antibiotics at that stage is also debatable [52]. The complex transmission pattern of leptospirosis involving more than 30 species of *Leptospira*, a wide range of reservoir animals, and the ecological conditions in tropical countries with abundant environmental exposure to water and wet soil makes it extremely difficult to control leptospirosis through source reduction. The only place of source reduction possible is in farming communities through extensive animal vaccination. However, animal vaccination strategies still need improvements in many settings [53,54]. Primary prevention through human vaccination seems more logical than source reduction, which also deserves consideration in highly exposed workers. Even though human vaccine studies were initiated early in the 20[th] century [55], effective pan-serovar vaccines are not available for human leptospirosis. The most recent work on virulent modification proteins of *Leptospira* species shows a possible candidate for a pan-serovar vaccine, yet the studies are still in the early stages [56,57].

Our estimates have several limitations. First, the productivity loss estimate was done using GDP, which tends to be an overestimation when the groups which are not economically active are also included in the sample. On the other hand, this study only provides productivity loss estimates, not the economic burden of leptospirosis. Direct costs, other indirect costs, and intangible costs are not included, and the cost of animal leptospirosis is not considered. The basic parameter used, DALYs, has many limitations [24,58] as an indicator of burden estimates. The DALYs estimated for leptospirosis also have severe limitations due to the use of national data in the estimation process [12]. Taken together, the actual global economic burden of leptospirosis could be much higher than estimated in our study. These numbers are estimated to worsen with the ecosystem dysfunctions due to climate change, especially with outbreaks of leptospirosis post floodings and cyclones of increasing frequency and intensity.

## Conclusions

Our study emphasises the substantial productivity cost of human leptospirosis, representing only a fraction of the overall economic impact. Estimating the economic impact of a disease requires a comprehensive assessment, considering direct medical costs, productivity losses due

to absenteeism and reduced labour efficiency, increased healthcare spending, and potential disruptions in supply chains and business operations. The main issue in the estimation of the economic burden of leptospirosis is the lack of primary data sources. The scarcity of field-based cost-of-illness (COI) studies will hamper the accuracy of any modelling exercises (econometrics, statistical models). We agree that it poses a significant challenge to the accuracy of modelling exercises, making it essential to urgently incorporate the COI assessment into field studies to comprehensively understand the global burden of leptospirosis. In addition to economic considerations, a thorough understanding of the disease epidemiology is essential to elucidate transmission patterns, affected population demographics, and potential long-term effects on workforce health. These considerations highlight the need for increasing the number of primary data points as the next step. Anticipated projections suggest an increase in leptospirosis cases attributable to unplanned urbanisation and climate change, demanding urgent action through a comprehensive One Health approach that integrates both animal and human health. Policymakers and health organisations are urged to prioritise leptospirosis as a Neglected Tropical Disease and allocate resources towards the development and implementation of effective prevention and control measures, including vaccine development.

## Supporting information

**S1 Table. Cost of leptospirosis due to loss of productivity by country.** The table includes all raw data used for the calculation and high and low estimates.
(XLSX)

## Author Contributions

**Conceptualization:** Suneth Agampodi, Jean-Louis Excler.

**Data curation:** Suneth Agampodi, Sajaan Gunarathna, Jung-Seok Lee, Jean-Louis Excler.

**Formal analysis:** Suneth Agampodi, Sajaan Gunarathna, Jung-Seok Lee, Jean-Louis Excler.

**Methodology:** Suneth Agampodi.

**Writing – original draft:** Suneth Agampodi.

**Writing – review & editing:** Suneth Agampodi, Sajaan Gunarathna, Jung-Seok Lee, Jean-Louis Excler.

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
