## [Decision Letter · Decision Letter 0]

21 Jul 2023

Dear Prof Agampodi,

Thank you very much for submitting your manuscript "Global, regional, and country-level cost of leptospirosis due to loss of productivity in humans" for consideration at PLOS Neglected Tropical Diseases. As with all papers reviewed by the journal, your manuscript was reviewed by members of the editorial board and by several independent reviewers. The reviewers appreciated the attention to an important topic. Based on the reviews, we are likely to accept this manuscript for publication, providing that you modify the manuscript according to the review recommendations. 

Sincerely,

Husain Poonawala

Academic Editor

Joseph Vinetz

Section Editor

Reviewer's Responses to Questions

**Key Review Criteria Required for Acceptance?**

**Methods**

-Are the objectives of the study clearly articulated with a clear testable hypothesis stated?

-Is the study design appropriate to address the stated objectives?

-Is the population clearly described and appropriate for the hypothesis being tested?

-Is the sample size sufficient to ensure adequate power to address the hypothesis being tested?

-Were correct statistical analysis used to support conclusions?

-Are there concerns about ethical or regulatory requirements being met?

Reviewer #1: Methods are retrospective but that limitation is inevitable given the data types available.

Reviewer #2: Appropriate

**Results**

-Does the analysis presented match the analysis plan?

-Are the results clearly and completely presented?

-Are the figures (Tables, Images) of sufficient quality for clarity?

Reviewer #1: Yes, the framework provided gives a uniquely important analysis that enables new insights into the global burden of human leptospirosis that complements previous DALY-based work.

Reviewer #2: Appropriate

**Conclusions**

-Are the conclusions supported by the data presented?

-Are the limitations of analysis clearly described?

-Do the authors discuss how these data can be helpful to advance our understanding of the topic under study?

-Is public health relevance addressed?

Reviewer #1: The conclusions are supported by the data presented and are very important to the field and beyond to global policy makers.

Reviewer #2: OK

**Editorial and Data Presentation Modifications?**

Reviewer #1: No real changes needed.

Reviewer #2: No

**Summary and General Comments**

Reviewer #1: I would ask the authors to comment on how they might suggest new study designs to provide the way to the next stage of analysis. For example, how would they present the present analysis to policy makers in LMIC countries where leptospirosis is endemic but the data are sparse?

Reviewer #2: The authors extensively estimate the cost of leptospirosis due to the loss of productivity in humans at the global, WHO regional, and country levels by converting the disability-adjusted life years (DALYs) lost due to leptospirosis into a monetary value using the per capita gross domestic product (GDP). Such estimation was not widely used in the past, but it is a meaningful way to convert DALYs (which may not make sense to many people and policymakers) into dollars (which is more understandable). The methodology employed in this study adheres to current standards and is straightforward. The results effectively represent different regions. Although the authors acknowledge several limitations of the estimation, at least there is an estimate of the monetary value of productivity loss due to leptospirosis.

Here is a comment regarding the manuscript:

Please provide further elaboration in the text manuscript about the concept of international dollars (Int$) in comparison to USD for better understanding and clarification.

PLOS authors have the option to publish the peer review history of their article (what does this mean?). If published, this will include your full peer review and any attached files.

Reviewer #1: No

Reviewer #2: No

Figure Files:

Data Requirements:

Reproducibility:

References

---

## [Editor Report · Decision Letter 1]

6 Aug 2023

Dear Prof Agampodi,

We are pleased to inform you that your manuscript 'Global, regional, and country-level cost of leptospirosis due to loss of productivity in humans' has been provisionally accepted for publication in PLOS Neglected Tropical Diseases.

Best regards,

Husain Poonawala

Academic Editor

Joseph Vinetz

Section Editor

---

## [Editor Report · Acceptance letter]

19 Aug 2023

Dear Prof Agampodi,

We are delighted to inform you that your manuscript, "Global, regional, and country-level cost of leptospirosis due to loss of productivity in humans," has been formally accepted for publication in PLOS Neglected Tropical Diseases.

Best regards,

Shaden Kamhawi

co-Editor-in-Chief

Paul Brindley

co-Editor-in-Chief
